# Clinical Relevance of Circulating Tumor Cells in Esophageal Cancer Detected by a Combined MACS Enrichment Method

**DOI:** 10.3390/cancers12030718

**Published:** 2020-03-18

**Authors:** Anna Woestemeier, Katharina Harms-Effenberger, Karl-F. Karstens, Leonie Konczalla, Tarik Ghadban, Faik G. Uzunoglu, Jakob R. Izbicki, Maximilian Bockhorn, Klaus Pantel, Matthias Reeh

**Affiliations:** 1Department of General, Visceral and Thoracic Surgery, University Medical Centre, Hamburg-Eppendorf, Martinistr. 52, 20246 Hamburg, Germany; a.woestemeier@uke.de (A.W.); k.karstens@uke.de (K.-F.K.); l.konczalla@uke.de (L.K.); t.ghadban@uke.de (T.G.); f.uzunoglu@uke.de (F.G.U.); izbicki@uke.de (J.R.I.); m.bockhorn@uke.de (M.B.); 2Department of Tumor Biology, University Medical Centre, Hamburg-Eppendorf, Martinistr. 52, 20246 Hamburg, Germany; k.effenberger@uke.de (K.H.-E.); pantel@uke.de (K.P.)

**Keywords:** esophageal cancer, circulating tumor cells, Ariol, CellSearch

## Abstract

Introduction. Current modalities to predict tumor recurrence and survival in esophageal cancer are insufficient. Even in lymph node-negative patients, a locoregional and distant relapse is common. Hence, more precise staging methods are needed. So far, only the CellSearch system was used to detect circulating tumor cells (CTC) with clinical relevance in esophageal cancer patients. Studies analyzing different CTC detection assays using advanced enrichment techniques to potentially increase the sensitivity are missing. Methods. In this single-center, prospective study, peripheral blood samples from 90 esophageal cancer patients were obtained preoperatively and analyzed for the presence of CTCs by Magnetic Cell Separation (MACS) enrichment (combined anti-cytokeratin and anti-epithelial cell adhesion molecules (EpCAM)), with subsequent immunocytochemical staining. Data were correlated with clinicopathological parameters and patient outcomes. Results. CTCs were detected in 25.6% (23/90) of the patients by combined cytokeratin/EpCAM enrichment (0–150 CTCs/7.5 mL). No significant correlation between histopathological parameters and CTC detection was found. Survival analysis revealed that the presence of more than two CTCs correlated with significantly shorter overall survival (OS) and progression-free survival (PFS). Conclusion. With the use of cytokeratin as an additional enrichment target, the CTC detection rate in esophageal cancer patients can be elevated and displays the heterogeneity of cytokeratin (CK) and EpCAM expression. The presence of >2CTCs correlated with a shorter relapse-free and overall survival in a univariate analysis, but not in a multivariate setting. Moreover, our results suggest that the CK7/8^+^/EpCAM^+^ or CK7/8^+^/EpCAM^−^ CTC subtype does not lead to an advanced tumor staging tool in non-metastatic esophageal cancer (EC) patients.

## 1. Introduction

Esophageal cancer (EC) is one of the most aggressive tumors with early metastatic spread and a five-year-survival rate of 18–25% after diagnosis [1]. Complete resection of the primary tumor with tumor-free margins and a multimodal treatment approach remains the only curative treatment of EC [2]. More than two thirds of all patients with esophageal cancer develop metastases or local recurrence, and circulating tumor cells in blood (CTCs) or disseminated tumor cells in bone marrow (DTCs) supposedly play a key part in this development [3,4,5]. Although, most of the shed tumor cells may die within the circulation, due to physical and anatomic conditions, some CTCs display stem cell characteristics and a malignant invasive potential to different organs, the lymphatic system, and importantly, blood and bone marrow [6,7]. Previous studies have demonstrated that the presence of CTCs correlated with shorter overall survival in patients with metastatic disease [5,8,9,10].

Within the last decade, many CTC detection techniques have evolved; they include, for instance, density gradient centrifugation [11,12], immunomagnetic cell enrichment [13,14], flow cytometry [15,16], filtration and isolation by size [17], PCR-based assays with various selected markers [14,18,19], or immunoassays against surface antigens (CellSearch, “CTC-chip”, flow cytometry) [20,21]. Most of these techniques are not implemented in clinical routine, due to a lack of standardization, reproducibility, or assay duration. Only the CellSearch system (Menarini Silicon Biosystems, Castel Maggiore, Italy, former Janssen Diagnostics, Raritan, NJ, USA) has been approved by the United States (U.S.) Food and Drug Administration for the use of reliable CTC detection in metastatic breast, colon, and prostatic cancers (K073338). Results obtained by this system have been demonstrated to be of prognostic significance in several tumor entities [22,23,24]. The detection is based on immunomagnetic enrichment for cells that express epithelial cell adhesion molecules (EpCAM) with antibody-coated magnetic beads and subsequent fluorescent staining for CTC evaluation [25]. EpCAM is frequently expressed on the surface of solid tumor cells; however, it has also been described to be down-regulated in CTC populations having undergone epithelial–mesenchymal transition [26]. Changes in the expression profile of disseminated tumor cells can lead to failure of EpCAM-based enrichment techniques for CTC detection [27]. Various groups also described the parallel expression of epithelial and mesenchymal markers on CTCs [28,29]. Epithelial markers such as cytokeratins (CK) are important filaments of the cytoskeleton and undergo changes of their expression profile during tumor progression from luminal CKs as CK8 or CK18 towards basal CKs such as CK5/6 or CK7 [30].

Therefore, CTC analysis was performed using anti-EpCAM, -CK7, and -CK8 as capture antibodies for Magnetic Cell Separation (MACS) enrichment (Miltenyi Biotech GmbH, Bergisch Gladbach, Germany) with subsequent immunocytochemical staining [31]. The objective of this study was to test whether the method increases the number of detected CTCs in esophageal cancer and is displaying a clinical impact in the prognosis of EC.

## 2. Materials and Methods

### 2.1. Study Design

In total, 90 initially resectable esophageal cancer patients, who underwent surgery at the University Hospital Hamburg-Eppendorf, were enrolled in this prospective study. Informed consent was obtained from all patients and the study was approved by the Medical Ethical Committee, Hamburg, Germany (PV3548). Only patients with histologically proven esophageal cancer were included in this study. None of the patients received perioperative treatment and all patients underwent Ivor–Lewis esophagectomy. Demographic, clinical, operative and postoperative data were gathered for each patient. Histopathological analysis was performed in accordance with the seventh edition of the tumor-node-metastasis (TNM) classification of the International Union against Cancer [32] by a senior specialist in gastrointestinal pathology. 

Postoperative follow-up was conducted at 3-month intervals for the first 2 years, and afterwards at 6-month intervals or until death. 

### 2.2. Tumor Cell Detection

Preoperatively, peripheral blood samples (7.5 mL) were collected in CellSave preservative tubes (Menarini Diagnostics, Berlin, Germany) and stored up to 72 h at room temperature until being processed. CTCs were detected by manual MACS enrichment (anti-cytokeratin 7 and 8 plus anti-EpCAM) with subsequent anti-cytokeratin staining, automated scanning, and evaluation by two trained scientists (Ariol SL-50™, Applied Imaging, Leica Microsystems, Wetzlar, Germany, thereafter referred to as “Ariol”).

For CTC enumeration, red blood cells were lysed (RBC Lysis buffers; Thermo Fisher Scientific, Boston, MA, USA), followed by the blocking of unspecific binding sites, cell permeation, and fixation in a one-step procedure (Carcinoma Cell Enrichment and Detection Kit; Miltenyi Biotec, Bergisch Gladbach, Germany) [31,33]. After incubation, the cells were centrifuged, resuspended in 1 mL PBS solution and applied to an MS separation column attached to an OctoMACS separator (both Miltenyi Biotec, Bergisch Gladbach, Germany). The columns were washed for three times with PBS solution and released from the separator. Target cells were eluted into 3–6 Hettich cytospin chambers (Hettich, Tuttlingen, Germany) and spun onto Poly-PrepTM PLL glass slides (Sigma, St. Louis, MO, USA) thereafter. The supernatant was removed and the slides were air-dried for 30 to 60 min. Target cells were fixed in 100% acetone (Merck, Darmstadt, Germany, former Sigma-Aldrich) at −20 °C for 10 min and the slides were dried at room temperature for 30 min. For CTC detection, three fluorescein isothiocyanate (FITC)-labeled, anti-cytokeratin antibodies were applied, namely AE1, AE3, and C11. Hematopoietic cells were stained with an anti-CD45 antibody directly labeled with DyeLight 549 (CTC Enrichment and Detection Kit, Genetix, New Milton, UK). A Dapi-containing mounting medium (Vector Laboratories, Burlingame, CA, USA) was used to visualize the nuclei. Automated slide scanning for FITC, Texas Red, Dapi, and brightfield was performed using the Ariol^®^ SL-50 system (Leica Biosystems, Wetzlar, Germany; CTC 3.3 software). The images of FITC (CK)-positive events were captured, presented by the system, and then evaluated by two trained scientists each. An event was classified as a tumor cell with the presence of a nucleus, cytokeratin expression, round or oval cell morphology, and absent CD45 expression (Figure 1).

The method was first described by Deng et al. analyzing the blood of breast cancer patients, therefore MCF-7 cells were spiked into blood, enriched, and stained as described above [31]. The recovery rate of these spiking experiments was above 90%. The staining served as an internal control in each run. Blood samples from healthy donors served as negative controls and were used to verify the specificity of the method. 

Detection of DTCs from bone marrow was previously described in detail [34].

### 2.3. Statistical Analysis

For statistical analysis PASW Statistics 18 (SPSS Inc., Chicago, IL, USA) was applied. Descriptive statistics were used to describe patient baseline characteristics. To evaluate a potential association between CTCs and clinicopathological parameters, the chi-square test was used. Events considered for survival analysis were death, local recurrence, and distant metastasis. When no events were recorded, the patients were censored at the last contact for statistical evaluation. Survival curves for progression-free (PFS) and overall survival (OS) of the patients were plotted by the Kaplan–Meier method and analyzed using the log-rank test. Results are presented as median survival in months with 95% confidence interval (CI) and number of patients at risk. Mean values are presented and specifically indicated in case the median survival was not reached. The OS was computed as the time from the date of surgery to either the date of death or last follow-up, whichever occurred first. The PFS was defined as the time from the date of surgery to the date of recurrence, last follow-up or date of death, whichever occurred first. Results are presented as hazard ratio with 95% confidence interval. Significant statements refer to *p*-values of two-tailed tests of less than 0.05. 

## 3. Results

### 3.1. Patient Characteristics and Tumor Cell Detection

Twenty-seven patients with squamous cell carcinoma (SCC), 60 with adenocarcinoma (AC), two patients with anaplastic carcinoma, and one patient with mixed SCC and AC of the esophagus were enrolled. The mean patient age was 63.7 years, 67 (74.4%) patients were male, and 23 (25.6%) were female.

Blood samples from 90 esophageal cancer patients were analyzed using a combined cytokeratin/EpCAM enrichment. CTCs were detected in 25.6% (23/90) of the patients with CTC numbers ranging from 0 to 150 (mean 2.6 CTCs). 

Moreover, the bone marrow of 63 of the 90 patients was analyzed and showed disseminated tumor cells (DTC) in 15.9% of the patients (10/63) (DTC count 0–6). There was an overlap in 5 patients showing CTCs and DTCs at the same time, but no significant correlation was found between the tumor cells in blood and bone marrow (*p* = 0.301, respectively; data not shown). 

### 3.2. Correlation with Histopathologic Parameters

Interestingly, neither the presence of one nor of >2 CTCs detected by MACS enrichment/Ariol correlated significantly with the histology (*p* = 0.771) or with routine histopathologic parameters (Table 1). 

### 3.3. Survival Analysis

The median survival time was 28 months. We were able to follow up on surviving patients for 38 months. Within the observation time, 65 of 90 patients showed tumor recurrence or died. 

In univariate survival analysis, patients with ≥1 CTC did not show significant differences in PFS (*p* = 0.053) and OS (*p* = 0.278) compared with CTC-negative patients (Figure 2). Using a cutoff value of >2 CTCs to be considered as CTC positive, patients with CTCs showed significantly shorter relapse-free (*p* = 0.020) and overall survival (*p* = 0.015) (Figure 3). 

Furthermore, in univariate analysis, OS was significantly influenced by the M stage (*p* = 0.001), the resection margins (0.02), and age (*p* = 0.001).

Histopathologic factors that turned out to be significant in univariate analysis, the pT stage, and lymph invasion were included into the multivariate model. While age (>60 vs. ≤60 years), metastatic stage, and pT stage were independent risk factors of overall survival, CTCs were not in this patient cohort (Table 2). 

## 4. Discussion

Preoperative staging in EC remains inaccurate, despite several diagnostic tools such as endoscopic ultrasound and computed tomography [35,36]. In the metastatic cascade, CTCs play an important role [37,38,39,40] and many different methods have been developed to enumerate tumor cells from whole blood samples. However, the CellSearch system is the only FDA-cleared system for CTC detection and it has been proven highly predictive of progression-free survival and overall survival in metastatic breast, colon, and prostatic cancers [22,23,24,41]. We recently confirmed these results by showing that CTCs detected by the CellSearch system are prognostic indicators of patients’ outcome in resectable esophageal cancer [9].

However, single immunomagnetic anti-EpCAM-based detection assays have major limitations and may underestimate the real number of CTCs in cancer patients. During tumor cell invasion and metastasis, some cells undergo epithelial to mesenchymal transition, and thus not all CTCs will express epithelial markers [27,42]. This heterogeneous expression and downregulation of EpCAM has been described in different cancer entities [26,28,29]. Therefore, tumor cells that express CK but low or no EpCAM may not be enriched by anti-EpCAM antibodies [31]. The use of anti-CK antibodies as an additional enrichment target could overcome this problem. So far, there is a lack of prospective studies analyzing different CTC detection assays in esophageal cancer patients. Therefore, this study, for the first time, investigated the clinical relevance of CTCs detected by the combined cytokeratin/EpCAM enrichment method in patients with resectable EC. The MACS enrichment/Ariol method was first described by Deng et al. [31] and can detect three types of CTCs: EpCAM^+^/CK^+^, EpCAM^−^/low/CK^+^, and EpCAM^+^/CK^−^/low, whereas the CellSearch system only detects EpCAM^+^ CTCs. Deng et al. analyzed blood samples of 49 patients with metastatic breast cancer using the combined enrichment assay and the CellSearch system. CTC detection rates were significantly higher with the MACS enrichment/Ariol (49% vs. 29%), and the authors concluded that anti-CK can be used for efficient CTC enrichment and increases the assay sensitivity. 

This study showed concordant results for EC patients: CTC detection rates were higher using MACS enrichment/Ariol with 25.6% compared with our previous study with 18.0% using CellSearch [9]. The mean number of CTCs varies widely in samples from different tumor entities [20]. In metastatic breast cancer and prostatic cancer, 5 or more CTCs showed an independent, prognostic, and predictive value [23,24], whereas the cutoff value in metastatic colon cancer is considered ≥3 CTCs [22]. Our study population mainly consists of patients defined as non-metastatic by preoperative staging. Within the course of disease, CTCs change their phenotype through epithelial-to-mesenchymal transition/mesenchymal-to-epithelial transition (EMT/MET) [43], potentially causing changes in the detection sensitivity of the technique applied. According to Allard et al., a cutoff level of >2 CTCs for MACS enrichment/Ariol was used in this study and showed significantly shorter relapse-free (*p* = 0.02) and overall survival (*p* = 0.015). Furthermore, given the single-center nature of this study, further prospective multi-center studies with larger patient cohorts are needed.

Moreover, DTCs in the bone marrow were detected in 15.9% (10/63) of the patients. DTCs can survive in the bone marrow in a dormant state for several years and might contribute to tumor recurrence by entering the blood stream again [39,44]. There was no significant correlation between the tumor cells in blood and bone marrow within our patient cohort, strengthening the hypothesis that these subtypes of isolated tumor cells in the body deliver distinct information about the course of the disease. 

In the multivariate analysis, the risk of a shortened OS was 1.9 times higher if CTCs were detected, but CTCs were not an independent prognostic marker (Table 2). This means that only EpCAM-enriched CTCs seem to serve as an independent prognostic biomarker for overall survival [9]. One may speculate that the higher CTC detection rate may lead to an increased significance in the prognostic value, but CTCs additionally enriched by CK7/8 do not seem to play a prognostic role for esophageal cancer patients, at least in our study cohort.

A caveat of EpCAM-based detection techniques is the morphology of the tumor and CTCs. It was shown that tumor cells spreading into the circulation may undergo phenotypic changes, known as epithelial to mesenchymal transition. In the metastatic process, some tumor cells lose their epithelial features and acquire a migratory mesenchymal phenotype. These cells are associated with a high malignancy potential and are postulated to be responsible for distant metastases and tumor relapse [28,45]. The MACS enrichment/Ariol as well as the CellSearch system might fail to detect all CTCs, especially those with high malignancy potential, due to a lack of epithelial markers [26]. Furthermore, using the MACS enrichment with prior permeabilization to EpCAM staining does not rule out the possibly of detecting cancer associated macrophage-like (CAMLs) cells. 

## 5. Conclusions

With the use of cytokeratin as an additional enrichment target, the CTC detection rate in esophageal cancer patients could be elevated. This displays the heterogeneity of CK and EpCAM expression. The presence of >2CTCs correlated with a shorter relapse-free and overall survival in a univariate analysis, but not in a multivariate setting. Since the main difference in our previous CTC study, which identified CTCs as an independent prognostic marker for EC patients, lies in the current technique enriching for CK7/8 in addition to EpCAM, our results lead to the hypothesis that the CK7/8^+^/EpCAM^+^ or CK7/8^+^/EpCAM^−^ CTC subtype does not lead to an advanced tumor staging tool in non-metastatic EC patients.

## Figures and Tables

**Figure 1 cancers-12-00718-f001:**
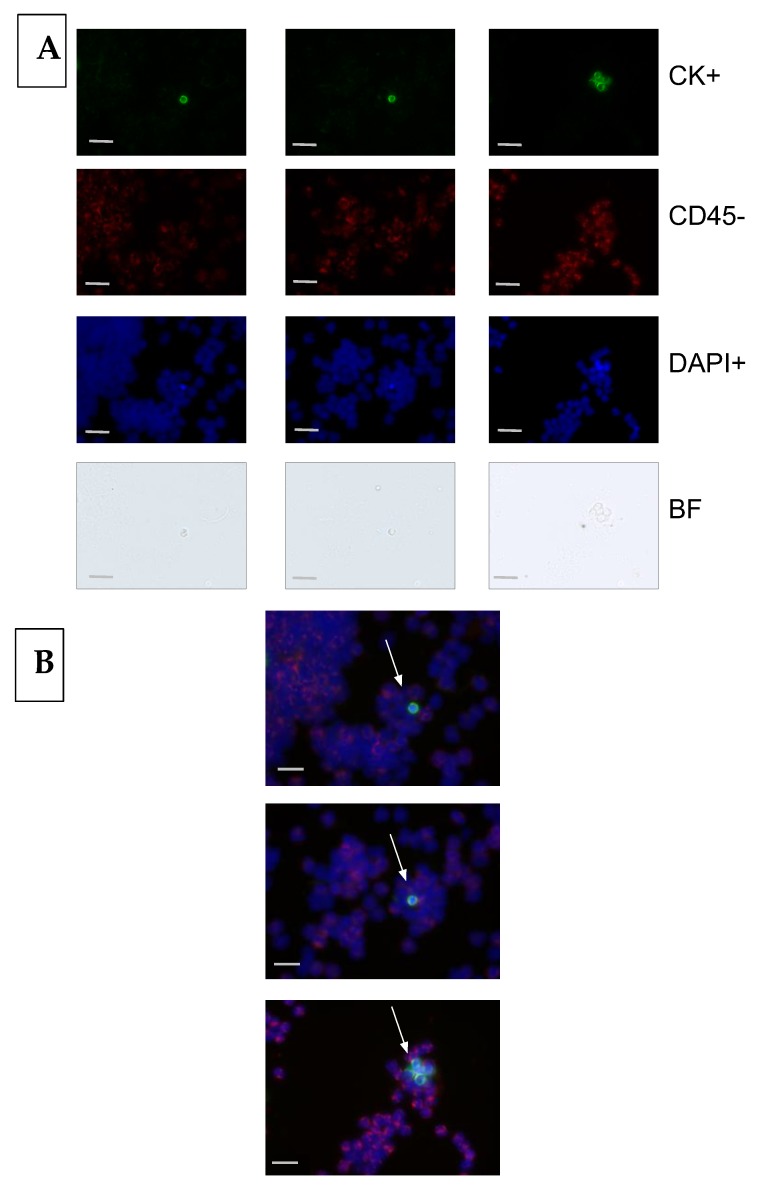
Image analysis of circulating tumor cells (CTCs) detected in the blood of esophageal cancer patients. Bar indicates 20 µm. (**A**) Cells classified as a CTC: positive for cytokeratin (CK), negative for CD45, positive for nuclear staining, and identified as an intact cell through the brightfield (BF) image. (**B**) Composite image of a CTC cluster from esophageal cancer patients.

**Figure 2 cancers-12-00718-f002:**
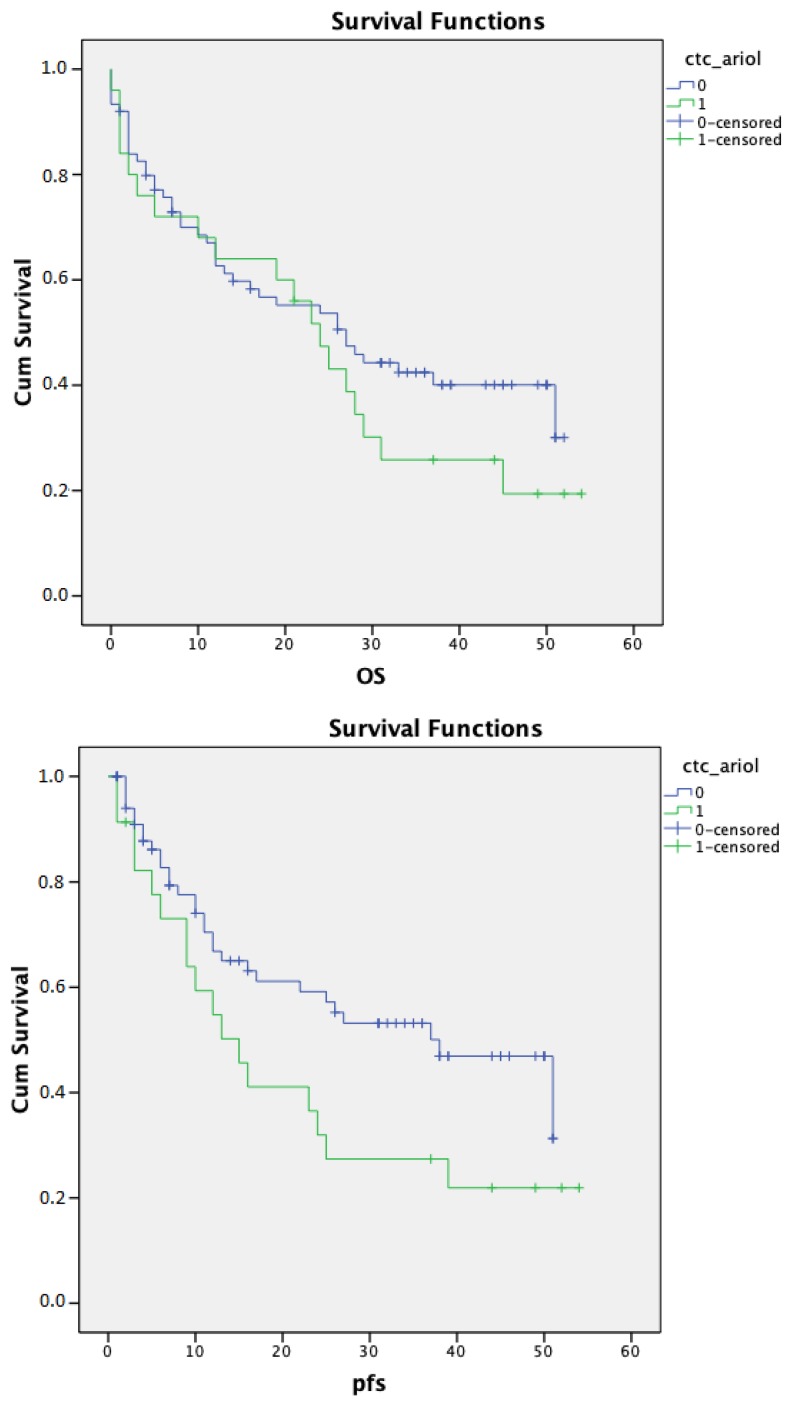
Kaplan–Meier curves for overall and progression-free survival according to ≥1 circulating tumor cell (CTC) detection by Magnetic Cell Separation (MACS) enrichment/Ariol. Patients with ≥1 CTC detected by MACS enrichment/Ariol did not show significantly shorter relapse-free (*p* = 0.053) and overall survival (*p* = 0.278) compared with CTC-negative patients.

**Figure 3 cancers-12-00718-f003:**
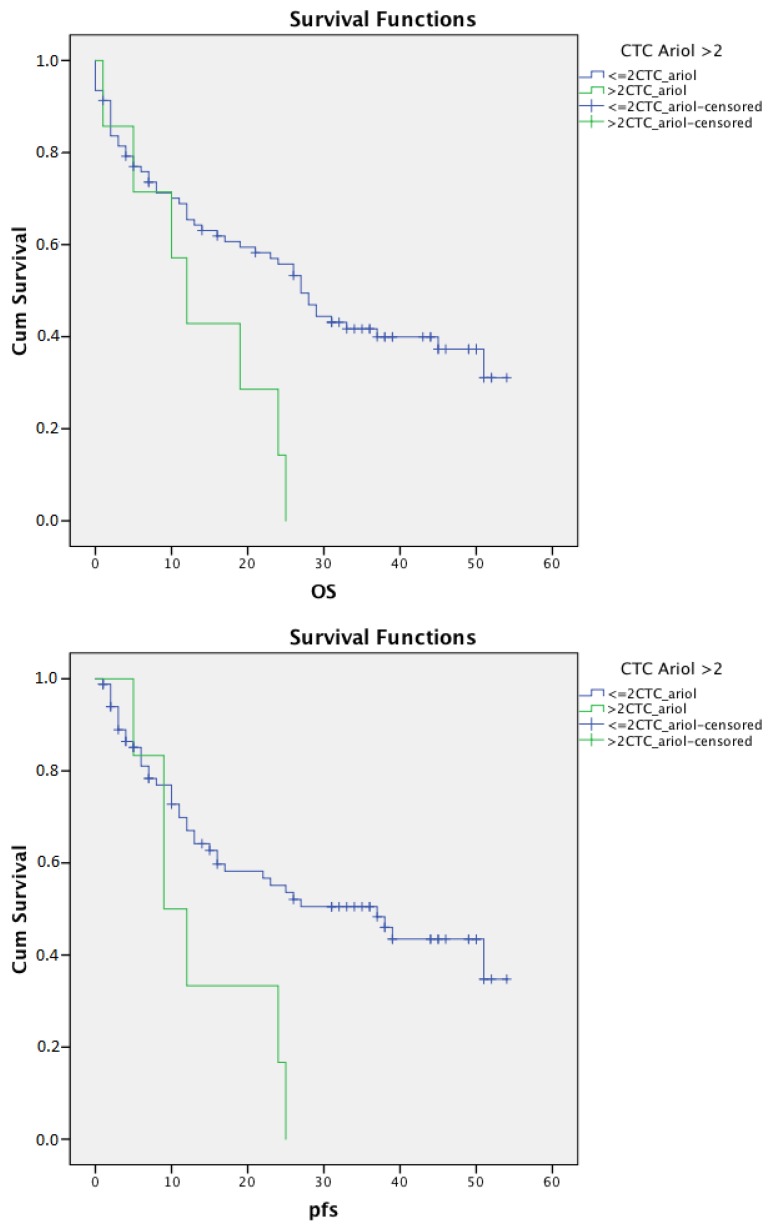
Kaplan-Meier curves for overall and progression-free survival according to >2 circulating tumor cells (CTC) detection by Magnetic Cell Separation (MACS) enrichment/Ariol. Patients with >2 CTCs detected by MACS enrichment/Ariol showed significantly shorter relapse-free (*p* = 0.02) and overall survival (*p* = 0.015) compared with CTC-negative patients.

**Table 1 cancers-12-00718-t001:** Correlation between histopathologic parameters and CTCs detected by Magnetic Cell Separation (MACS) enrichment/Ariol. Missing refers to the respective test parameter.

Parameter	Patients (n = 90)	>2 CTC Ariol (%)	*p*-Value
Sex			0.519
Male	69	5 (7.2%)	
Female	21	2 (9.5%)	
pT stage			0.469
T1	22	2 (9.1%)	
T2	18	1 (5.6%)	
T3	36	2 (5.6%)	
T4	14	2 (14.3%)	
pN stage			0.320
N0	42		
N1	18	3 (7.1%)	
N2	14	2 (11.1%)	
N3	13	1 (7.1%)	
Missing	3	0	
M stage			0.284
M0	85	6 (7.1%)	
M1	4	1 (25.0%)	
Missing	1		
UICC stage			0.327
0	2	0	
I	24	1 (4.2%)	
II	16	2 (12.5%)	
III	39	3 (7.7%)	
IV	9	1 (11.1%)	
Tumor grading			0.360
G1–2	45	4 (8.9%)	
G3–4	43	2 (4.7%)	
Missing	2		
Resection margins			0.530
R0	77	5 (6.5%)	
R1	10	1 (10.0%)	
Missing	3		

**Table 2 cancers-12-00718-t002:** Multivariate analysis of overall survival. HR: hazard ratio; CI: confidence interval; CTC: circulating tumor cell.

Factor	HR	95% CI	*p*-Value
Ariol: >2CTC vs. ≤2 CTC	0.723	0.297–1.756	0.474
Age: >60 vs. ≤60 years	0.374	0.199–0.705	0.002
pT stage: pT1/2 vs. pT3/4	0.409	0.220–0.760	0.005
Lymph invasion: L0 vs. L1	0.771	0.435–1.365	0.372
M stage: M0 vs. M1	0.112	0.25–0.499	0.004
Resection margins: R0 vs. R1	0.770	0.374–1.583	0.477

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
