# Peer review of "Clinical Relevance of Circulating Tumor Cells in Esophageal Cancer Detected by a Combined MACS Enrichment Method"

_cancers, 2020, doi:10.3390/cancers12030718_

Round 1
Reviewer 1 Report
The paper is well written and presented. My only comments are in the conclusions. The authors stained only with cytokeratin and EpCAM. In order to allow cytokeratin antibodies inside the cell, the cells were permeablized, this allowed both cytokeratin and EpCAM antibodies inside. The CD45 in the 4 cell clump looks weakly positive in the CD45 field alone. It would have been nice to see a positive WBC control. Macrophages can be very weak in CD45 staining and macrophages can engulf tumor cells prior to free tumor cells being seen in the circulation. These macrophages have been referred to as cell associated macrophage-like (CAMLs). These cells cannot be counted as a free cancer cell. They have been seen in individuals before cancer has been diagnosed and have been identified around solid tumors. In order to rule out that these are not CAMLs the initial capture-staining with EpCAM should have been done prior to permeablization to only include cells with surface EpCAM staining.
Line 119 - add the definition of DTC.
Line 171 - change lymphinvasion to lymph invasion
The purpose of having the dual capture with cytokeratin and EpCAM was to guard against missing a CTC with loss of EpCAM; was AE1 and AE3 cytokeratin only added in the staining or was it added in the capture stage? I mention this because AE1 and AE3 also stain cytokeratin in fibroblasts which are numerous in the circulation. C11 is a broad spectrum cytokeratin mixture and should have been sufficient, it does lacks CK7 and CK18. It was not mentioned what cytokeratin was used in the capture.
Line 221 states the study population consisted mainly of non-metastatic patients. The finding of a true CTC in the circulation means the disease is metastatic by definition. Again CAMLs are seen in non-metastatic patients and sometimes prior to the detection of any cancer.
Author Response
Reviewer #1
- The paper is well written and presented. My only comments are in the conclusions. The authors stained only with cytokeratin and EpCAM. In order to allow cytokeratin antibodies inside the cell, the cells were permeablized, this allowed both cytokeratin and EpCAM antibodies inside. The CD45 in the 4 cell clump looks weakly positive in the CD45 field alone. It would have been nice to see a positive WBC control. Macrophages can be very weak in CD45 staining and macrophages can engulf tumor cells prior to free tumor cells being seen in the circulation. These macrophages have been referred to as cell associated macrophage-like (CAMLs). These cells cannot be counted as a free cancer cell. They have been seen in individuals before cancer has been diagnosed and have been identified around solid tumors. In order to rule out that these are not CAMLs the initial capture-staining with EpCAM should have been done prior to permeablization to only include cells with surface EpCAM staining.
Thank you for this valuable note. Unfortunately, we do not have an image of a positive WBC control for this sample. However, we discuss the possibility of detecting CAMLs instead of circulating tumor cells in lines 250-252.
- Line 119 - add the definition of DTC.
The definition of DTC is mentioned before in lines 40,41: ´disseminated tumor cells in the bone marrow´.
- Line 171 - change lymphinvasion to lymph invasion
Lymphinvasion was changed to lymph invasion.
- The purpose of having the dual capture with cytokeratin and EpCAM was to guard against missing a CTC with loss of EpCAM; was AE1 and AE3 cytokeratin only added in the staining or was it added in the capture stage? I mention this because AE1 and AE3 also stain cytokeratin in fibroblasts which are numerous in the circulation. C11 is a broad spectrum cytokeratin mixture and should have been sufficient, it does lacks CK7 and CK18. It was not mentioned what cytokeratin was used in the capture.
Anti-cytokeratin 7 and 8 were added in the capture stage. This is mentioned in line 70 and we added this to the materials and methods part (line 91).
- Line 221 states the study population consisted mainly of non-metastatic patients. The finding of a true CTC in the circulation means the disease is metastatic by definition. Again CAMLs are seen in non-metastatic patients and sometimes prior to the detection of any cancer.
We modified this section to ´consists of patients defined as non-metastatic by preoperative staging ´. Indicating that the patients staged as non-metastatic by CT- scan, sonography and endoscopy are indeed metastatic by definition (lines 225)
Reviewer 2 Report
While the methodology of this of the study is interesting, I have a few concerns:
It is not clear from the manuscript if the CTC detection rate was influenced by histology (adeno vs SCC). Treatment details are not available, therefore any conclusions regarding the correlation between CTC and survival seem premature at this point. The generalizability of the results is questionable given the single-center nature of the study.Author Response
Reviewer #2:
While the methodology of this of the study is interesting, I have a few concerns:
- It is not clear from the manuscript if the CTC detection rate was influenced by histology (adeno vs SCC).
The CTC detection rate was not influenced by histology. We specified this in the manuscript in line 159.
- Treatment details are not available, therefore any conclusions regarding the correlation between CTC and survival seem premature at this point.
In this study, only samples obtained at diagnosis before treatment were
collected. A caveat of this study is, that postoperative treatment details were not considered in the survival analysis.
However, large, prospective, multicenter studies have validated CTC detection by the cellsearch system as the strongest prognostic factor for overall and recurrence-free survival. (Cristofanilli M et al. J Clin Oncol. 2005; Cohen SJ et al. Ann Oncol. 2009; Reeh M et al. Ann Surg. 2015).
- The generalizability of the results is questionable given the single-center nature of the study.
The limitation of this single-center study is added to lines 229, 230: ´Furthermore, given the single-center nature of this study, further prospective multi-center studies with larger patient cohorts are needed.´
Reviewer 3 Report
The CellSearch system is intended for the enumeration of circulating tumor cells (CTC) enriched by EpCAM, with the subtype of EpCAM+, CD45-, and cytokeratins 8, 18+, and/or 19+ in whole blood. The limitation of CellSearch is the underestimation of CTCs undergoing epethilial to mesenchymal transition (EMT) or CTCs from non-epithelial origin. In this study, other than EpCAM, the cytokeratin was added as additional marker to enrich the CTCs. Although the proposed method MACS/Ariol increases the sensitivity for CTCs detection, there are still some questions/concerns (see the following) need to be addressed before publication.
1) There is a need for extensive prospective evaluation based on a sizeable number of clinical samples, this study is based on 90 esophageal patients, the sample number is not enough;
2) Although, CTCs detection based on additional cytokeratin biomarkers elevated the assay sensitivity, the CTCs additionally enriched by CK7/8 do not seem to play prognostic role for esophageal cancer patients, which is unlike the EpCAM-enriched CTCs by CellSearch; The CK7/8+ EpCAM+, CK7/8+/EpCAM- CTC subtype does not lead to advanced tumor staging for non-metastatic EC patients. Does that mean the CTCs detection based on MACS system might not lead to increased significance in the prognostic value?
3) In line 115, it’s unclear why the breast cancer cell line MCF-7 was selected as a model cell line for the esophageal cancer study?
4) In line 163, “Within the observation time, 65 of 100 patients showed tumor recurrence or died.” Why 100 patients? As only 90 patients were enrolled in your study.
Author Response
Reviewer #3:
The CellSearch system is intended for the enumeration of circulating tumor cells (CTC) enriched by EpCAM, with the subtype of EpCAM+, CD45-, and cytokeratins 8, 18+, and/or 19+ in whole blood. The limitation of CellSearch is the underestimation of CTCs undergoing epethilial to mesenchymal transition (EMT) or CTCs from non-epithelial origin. In this study, other than EpCAM, the cytokeratin was added as additional marker to enrich the CTCs. Although the proposed method MACS/Ariol increases the sensitivity for CTCs detection, there are still some questions/concerns (see the following) need to be addressed before publication.
- There is a need for extensive prospective evaluation based on a sizeable number of clinical samples, this study is based on 90 esophageal patients, the sample number is not enough;
The limitation of this single-center study is added to lines 229, 230.
- Although, CTCs detection based on additional cytokeratin biomarkers elevated the assay sensitivity, the CTCs additionally enriched by CK7/8 do not seem to play prognostic role for esophageal cancer patients, which is unlike the EpCAM-enriched CTCs by CellSearch; The CK7/8+ EpCAM+, CK7/8+/EpCAM- CTC subtype does not lead to advanced tumor staging for non-metastatic EC patients. Does that mean the CTCs detection based on MACS system might not lead to increased significance in the prognostic value?
Yes, the MACS enrichment (anti-cytokeratin 7 and 8 plus anti-EpCAM) increases the CTC detection rate, but does not increase the prognostic value:
´Our results lead to the hypothesis that the CK7/8+/EpCAM+ or CK7/8+/EpCAM- CTC subtype does not lead to an advanced tumor staging tool in non-metastatic EC patients´ (lines 261- 263).
- In line 115, it’s unclear why the breast cancer cell line MCF-7 was selected as a model cell line for the esophageal cancer study?
Since the MACS enrichment/ Ariol method was first described by Deng et al. 2008 and
analyzed blood samples of patients with metastatic breast cancer, the breast cancer cell line MCF-7 served as an internal control in our study.
The explanation was added in lines 115-117.
- In line 163, “Within the observation time, 65 of 100 patients showed tumor recurrence or died.” Why 100 patients? As only 90 patients were enrolled in your study.
We apologize, only 90 patients were enrolled. We corrected the sentence (line 167).
Round 2
Reviewer 2 Report
Thanks to the authors for the revisions. I still think the study is limited because of the small sample size and the fact that treatment details are not available.
Author Response
Thanks to the authors for the revisions. I still think the study is limited because of the small sample size and the fact that treatment details are not available.
We thank reviewer 2 for the comments.
We included nearly 100 patients with esophageal cancer which in our opinion is an adequate and sufficient cohort size compared to other reports published in good journals.
Concerning the second comment of reviewer 2, we included the treatment details in the methods part: “None of the patients received perioperative treatment and all patients underwent Ivor-Lewis esophagectomy.” (see page 2, line 81-82)
Reviewer 3 Report
The authors have addressed all the comments and the paper is publishable after typo and grammar checking.
Author Response
The authors have addressed all the comments and the paper is publishable after typo and grammar checking.
We thank reviewer 3 for this positive feedback. We performed typo and grammar checking and hope that the manuscript is now suitable for publication.
Round 3
Reviewer 2 Report
I would like to thank the authors for their response.